# Robust Gas-Path Fault Diagnosis with Sliding Mode Applied in Aero-Engine Distributed Control System

Xiaodong Chang  and Xiaojie Qiu *

AECC Aero Engine Control System Institute, Wuxi 214063, China; cxd18762406179@163.com
* Correspondence: qzhang6008@126.com; Tel.: +86-187-6240-6179

**Abstract:** The technology of aero-engine gas-path fault diagnosis is an important way to improve flight safety and reliability and reduce maintenance costs. With the maturity of the new-generation engine distributed control system (DCS), uncertainties such as bus packet loss, time delay, and node function degradation have increasingly highlighted new challenges to engine fault diagnosis. At present, linear Kalman filter (LKF) is widely researched and used in engine fault detection and isolation (FDI), but its robustness has proved to be not strong. However, the sliding mode observer (SMO) is not only capable of fault reconstruction but also robust to system uncertainties and disturbances due to its unique discontinuous switching term, which tends to be an effective way to achieve robust fault diagnosis for aero engines and DCS with many uncertainties. This paper initially develops a distributed bus packaging model that supports time-delay and packet-loss simulating and timing planning based on SimEvents, providing a basis for the model-based design and verification. Then the SMO is adopted to design a robust gas-path diagnosis method for engine DCS, and the robust observing accuracy is improved by combining high-order sliding mode theory, LMI optimized observation matrix, and variable gain. The simulation results show the effectiveness and advantages in engine DCS application scenarios.

**Keywords:** aero-engine; distributed control system; bus; sliding mode observer; uncertainties

## 1. Introduction

The safety and reliability of aero-engines is an important premise to ensure flight safety. Once fault or failure occurs, it may not only mean huge economic losses but even lead to major catastrophic accidents. At present, aero-engine maintenance is not only concerned with accident prevention but gradually emphasizes the improvement of performance reliability. The maintenance methods are also transitioned from post-maintenance, timed maintenance to condition based maintenance (CBM) based on real-time monitoring. The performance monitoring technology is used to monitor engine performance parameters in real time or regularly, so as to optimize the maintenance timing and schedule, which not only ensures engine reliability, but also greatly saves operating costs [1,2].

Over repeated operations, the gas-path performance of aero-engines gradually deteriorates. Common causes of gradual degradations include the compressor fouling, increase in the blade-tip clearance in the turbine, labyrinth seal leakage, wear and erosion, and corrosion in the hot sections. In addition, foreign-object damage, caused by impingement of such foreign-objects as birds, pieces of ice, and runaway debris, will cause abrupt performance shifts. The variations of efficiency and flow capacity of gas-path components, called "health parameters", capture the nature of engine performance. They deviate from the nominal baseline gradually with time as engine parts wear from regular usage, and also abruptly due to component fault events. The health parameters cannot be directly measured during the flight, but fortunately, their degradations cause changes in the observable parameters, such as temperature, pressure, and rotational speed. Additionally, sensors in aircraft engines operate in severely hostile conditions, thus they are prone to faults and failures. Any

undetected sensor faults may cause disastrous consequences to engine control loops, and even threaten flight safety. Therefore, the evaluation of engine performance based on the health parameters estimation and sensor fault detection helps operators determine the regular maintenance schedule and arrange replacement of components when performance reaches unacceptable levels.

Under this background, an engine health management system (EHM) has been put forward [3,4]. EHM is a real-time management system integrating the latest engine fault detection, analysis, and diagnosis technology, which is the key technology to realize the CBM of aero-engine. In the EHM system, gas path fault diagnosis technology plays an important role [5]. Gas path fault diagnosis technology uses the parameter information collected by the engine gas path sensor, combined with the filter, intelligent algorithm, and other ways to analyze the current health condition of the engine gas path, and detects and locates faults. Typical fault diagnosis algorithms include analytic model-based [6–8] methods (least square method, Kalman filter) and data-driven [9–11] methods (neural network, wavelet analysis, fuzzy algorithm). Each of these methods has its own advantages and disadvantages, but the model-based gas-path fault diagnosis method is based on the physical equation reflecting the aero-thermodynamic properties of the engine to build a model, and taking engine faults into the model can achieve more accurate quantitative performance monitoring, and it has been widely concerned and studied.

For the aero-engine, applying distributed control systems [12] in the future, the current gas-path fault diagnosis methods mainly have the following problems: First, since the engine is a complex and variable strong nonlinear system, the modeling error is inevitable, and the distributed system causes the delay and packet loss; therefore, the diagnosis system faces strong uncertainty. How to achieve robust estimation and diagnosis needs to be resolved in model-based approaches. Second, traditional filter/observer methods regard the performance parameter as a state, which needs to assume that its derivative is zero. This assumption is obviously contrary to the fact of fast time-varying faults, which leads to the problem of slow convergence of the estimator when estimating abrupt fault cases. Third, aero-engines are often maneuvered in a wide flight range, while the existing diagnostic methods are mostly based on steady-state data. This is because the engine data in the transition state changes greatly, which requires high diagnostic robustness, but it is difficult to meet in existing methods.

In recent years, the application of sliding mode observer (SMO) in fault diagnosis has been widely studied. This is due to the fault reconstruction capability of SMO and its unique discontinuous switching terms, which ensure the robustness of the system to model uncertainties and disturbances. There are many researches and applications of sliding mode observer in the fault diagnosis of motor [13], ship [14], robot [15], and aircraft control systems [16]. In the field of aviation, Alwi [17,18] designed a robust sliding mode observer for actuator faults of aircraft control systems, and realized the fault diagnosis within the full envelope range of aircraft by extending the sliding mode observer to the LPV (linear parameter varying) model. Nader [19] designed an adaptive sliding mode observer for fault diagnosis of gas turbine sensors, where the degraded performance was taken into account, and accurate fault reconstruction can be realized throughout the lifespan. Loza [16] designed a non-homogeneous high-order sliding mode observer for sensor fault diagnosis of transport aircraft with good simulation results. Edwards [20] designed a sensor fault diagnosis system for civil aircraft by using the sliding mode observer and verified it on the flight simulator of an Airbus. At present, the application of the sliding mode observer is mainly focused on the aircraft control/diagnosis system, and the research on the aero-engine gas-path fault diagnosis can be rarely found. Previously, our research team [21,22] proposed a robust diagnosis method based on unknown input reconstructed sliding mode observer for aero-engine gas-path diagnosis, which could realize real-time fault reconstruction by using fault information contained in discontinuous switching terms. The assumption in LKF that the derivative of reconstructed signals is zero is avoided, which improves the diagnosis speed and robust estimation accuracy.

However, the application of sliding mode observer technology to the robust diagnosis of aero-engine DCS is rarely reported. In fact, the robust fault diagnosis capability of the sliding mode observer is an effective way to achieve a robust estimation of gas-path performance and robust fault diagnosis for aero-engine DCS with a lot of uncertainties. At present, one of the bottlenecks restricting the engineering application of gas path diagnosis technology is that the system uncertainty restricts the technical reliability. Therefore, the research of distributed system robust diagnosis technology based on sliding mode observer is of great practical significance to solve the core problem and promote the application of aero-engine DCS diagnosis.

## 2. Aero Engine Dynamic Model

The discussed engine plant in this paper is a dual-shaft separated-exhaust turbofan engine with a high bypass ratio. The variable stator vane angle ($\theta_{\mathrm{VSV}}$) and the variable bleed valve angle ($\theta_{\mathrm{VBV}}$) are adjustable. The main components and structural diagram are shown in Figure 1. The notations used in this paper and their descriptions are shown in Table 1.

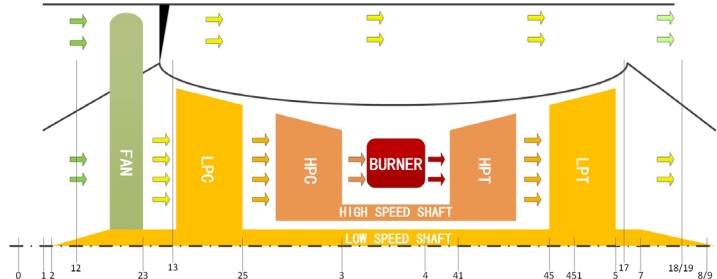

**Figure 1.** The structure of a dual-shaft separated-exhaust turbofan engine.

**Table 1.** The notations and their descriptions.

| Notation | Description |
| --- | --- |
| $H$ | Height |
| $M_{\mathrm{a}}$ | Mach number |
| $N_{\mathrm{L}}$ | Low pressure rotor speed |
| $N_{\mathrm{H}}$ | High pressure rotor speed |
| $\boldsymbol{h}$ | Health parameter vector |
| $SE1\ (h_1)$ | LPC efficiency |
| $SE2\ (h_2)$ | HPC efficiency |
| $SE3\ (h_3)$ | HPT efficiency |
| $SE4\ (h_4)$ | LPT efficiency |
| $SW1\ (h_5)$ | LPC flow capacity |
| $SW2\ (h_6)$ | HPC flow capacity |
| $SW3\ (h_7)$ | HPT flow capacity |
| $SW4\ (h_8)$ | LPT flow capacity |
| $\boldsymbol{f}$ | Sensor fault |
| $W_{\mathrm{f}}$ | Fuel flow rate |
| $\theta_{\mathrm{VSV}}$ | Variable stator vane angle |
| $\theta_{\mathrm{VBV}}$ | Variable bleed valve angle |
| $P_{25}$ | HPC inlet pressure |
| $T_{25}$ | HPC inlet temperature |
| $P_3$ | Combustor inlet pressure |
| $T_3$ | Combustor inlet temperature |
| $T_{495}$ | Exhaust gas temperature |

In this paper, the nonlinear mathematical model of the engine is established by the component analytic method, and the nonlinear system can be expressed as:

$$
\begin{aligned}
\dot{x} &= f(x, u, v) \\
y &= g(x, u, v)
\end{aligned} \tag{1}
$$

where $x \in R^n$ is the state and $y \in R^p$ is the output. $u \in R^m$ is the control input and $v$ denotes the external parameters (flight condition). The function $f$ and $g$ are, respectively, the engine process and measurement expressions. In the discussed engine plant, $x = [N_L, N_H]^T$, $y = [N_L, N_H, P_{25}, T_{25}, P_3, T_3, T_{495}]^T$ and $u = [W_f, \theta_{VSV}, \theta_{VBV}]^T$.

The health condition of engine gas-path components (abrupt fault or gradual performance degradation) can be characterized by component performance parameters. Therefore, the health parameter $h$ is introduced to quantify the degree of sudden and gradual performance degeneration of each component after fault occurs. Since the inner ducted fan is driven by the same shaft with and the LPC, its function is equivalent to the first stage of the LPC. In the component-level model, the inner ducted fan and the low-pressure compressor are often modeled as one component, while the bypass fan is modeled as a separate component. Since all seven sensors involved in this paper are located in the inner duct of the engine, the health parameters of the outer bypass are not considered. In addition, the combustion chamber efficiency generally varies little during the engine service life, therefore it is also not considered. Finally, the health parameter $h$ is specifically selected as the efficiency variation coefficient $SE$ and the flow variation coefficient $SW$ of the rotating component, which is defined as follows:

$$
SE_i = \frac{\eta_i}{\eta_i^*}, SW_i = \frac{W_i}{W_i^*} \tag{2}
$$

where $\eta_i, w_i$ are the actual efficiency and flow rate of the component, while $\eta_i^*, w_i^*$ are the corresponding ideal values, and the subscript $i$ ($i$ = 1, 2, 3, 4) represents the number of the component. Limited by the positions and quantities of actual engineering sensors, health parameters have to be selected [19,20]. In this paper, the flow rate of the low-pressure turbine is abandoned because there are no LPT-related sensors, and the health parameter vector is chosen as $h$ = $[SE_1 \ SW_1 \ SE_2 \ SW_2 \ SE_3 \ SW_3 \ SE_4]^T$.

In this paper, the Newton–Raphson (N–R) method is adopted to obtain the component-level model (CLM) of the aero-engine. The modeling technique of the CLM has been relatively mature, and the calculation formula of each component can be referred to in the literature [23]. Since the nonlinear model is seldom used in the design of control law and fault diagnosis algorithm and the linear algorithm has been successfully applied to practical control and diagnosis problems, a practical design method is to design linear control law (or diagnostic filter) for steady-state points, and then add nonlinear features, such as gain scheduling and switching, so that it can be applied to full envelope and state. For various linear control and diagnosis algorithms, the linear state variable model (SVM) is the basis of their design. Taylor expansion at a steady point $(x_0, u_0, y_0)$ of the engine and retaining constant and first-order terms yields the following SVM:

$$
\begin{aligned}
\dot{x} &= f(x, u) \approx f(x_0, u_0) + \frac{\partial f}{\partial x}\bigg|_{(x_0,u_0)} \Delta x + \frac{\partial f}{\partial u}\bigg|_{(x_0,u_0)} \Delta u \\
y &= g(x, u) \approx g(x_0, u_0) + \frac{\partial g}{\partial x}\bigg|_{(x_0,u_0)} \Delta x + \frac{\partial g}{\partial u}\bigg|_{(x_0,u_0)} \Delta u
\end{aligned} \tag{3}
$$

and Equation (3) can be further depicted as

$$
\begin{aligned}
\Delta \dot{x} &= A\Delta x + B\Delta u \\
\Delta y &= C\Delta x + D\Delta u
\end{aligned} \tag{4}
$$

where $A \in \mathbb{R}^{n \times n}, B \in \mathbb{R}^{n \times m}, C \in \mathbb{R}^{p \times n}, D \in \mathbb{R}^{p \times m}$ are the system matrices with appropriate dimensions. Here, $n = 2, m = 3, p = 7$ and $\Delta x = x - x_0, \Delta y = y - y_0, \Delta u = u - u_0$. For simplicity, the sign "$\Delta$" in Equation (4) is omitted in the following deductions. In this paper, a hybrid fitting method is applied to obtain the SVM, which is depicted in [24].

### 3. Integrated Model of Engine Distributed Control System

The discussed architecture of the engine control system is a partial DCS with high bandwidth and high fault tolerance TTP/C bus. The smart nodes (SN) include engine electronic central controller (EECC), front data concentrator (FDC), rear data concentrator (RDC), smart main fuel pump (MFP), smart actuator, etc., which are all in dual redundancy in hardware and software. Sensor signals are acquired, converted, filtered, and diagnosed by FDC and RDC, then information flow into EECC to calculate the control schemes and demands, which are the inputs of pump and actuators. Finally, currents are exported by controllers affiliated to pump and actuators to realize the closed-loop control (Figure 2).

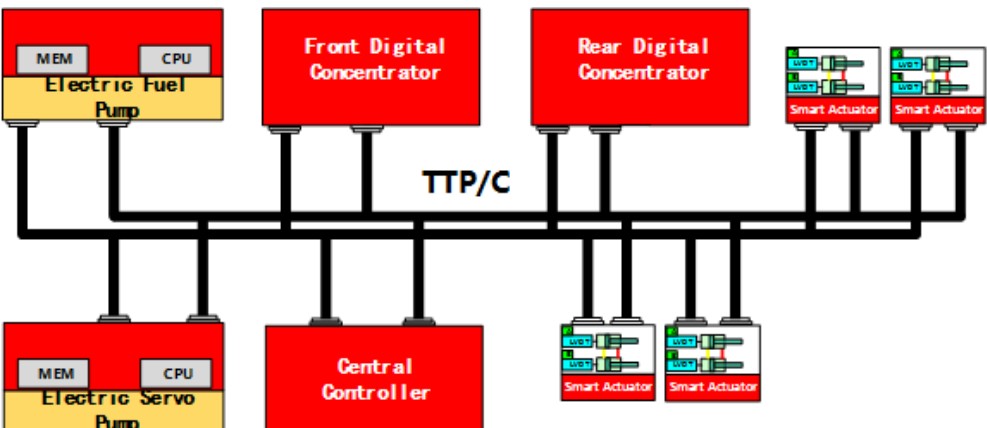

**Figure 2.** The architecture of engine distributed control system.

Compared with the traditional centralized control system, the distributed control system of aero-engines has the characteristics of distributed control functions, intelligent nodes, and a lightweight system. It significantly reduces the system coupling, total weight, and EECC calculating demands, and improves the anti-pollution, high-temperature resistance, anti-interference capability of accessories, system maintainability, and reliability, while also providing the basis for the application of advanced control technology. However, the existing aero-engine control system modeling methods cannot meet the requirements of DCS research. Since in DCS, control programs and data flow programs are implemented asynchronously, and system synchronization is achieved through the bus network; therefore, control system data flow is more complex, and system design needs to comprehensively evaluate the bus performance, timing scheduling, delay packet loss, and other influences on the system control performance.

Figure 3 describes the fundamental principle of TTP/C bus protocol. The TTP/C bus is a kind of time-triggered protocol, featured with TDMA (time division multiple access). The physical bus has dual channels in architecture, and any SN in the network broadcast its information to each channel. A TTP SN includes a TTP controller, a host controller (including AS and OS), and a TTP CNI (container network interface). The MEDL information is stored in a TTP controller, which contains information of the cluster cycle, TDMA round, node slot allocation, and all scheduling in the network.

In order to deeply study the distributed control system of the aero-engine, it is necessary to study the integrated modeling method. Although the Truetime toolbox [25] is a bus-specific simulation tool, its scalability is poor, and it is not suitable for system simulation including complex timing planning. Moreover, it does not support fixed-step simulation, and it is not suitable for engine environment modeling. SimEvents® toolbox (Matlab SimEvents R2017a) is a discrete event system modeling and simulation tool in Simulink® (Matlab Simulink R2017a), in which the bus signal flow can be simulated through "entity" to build a distributed network model so that simulating the bus network characteristics in DCS can be realized. Using SimEvents can solve the problems such as time-triggered TTP/C bus simulation and random bus fault injection simulation,

and realize the system-integrated simulation of engine control, system timing, and distributed bus behavior. In addition, it supports bus fault random injection during operation, which provides convenience for function design, timing design, and security analysis of distributed architecture.

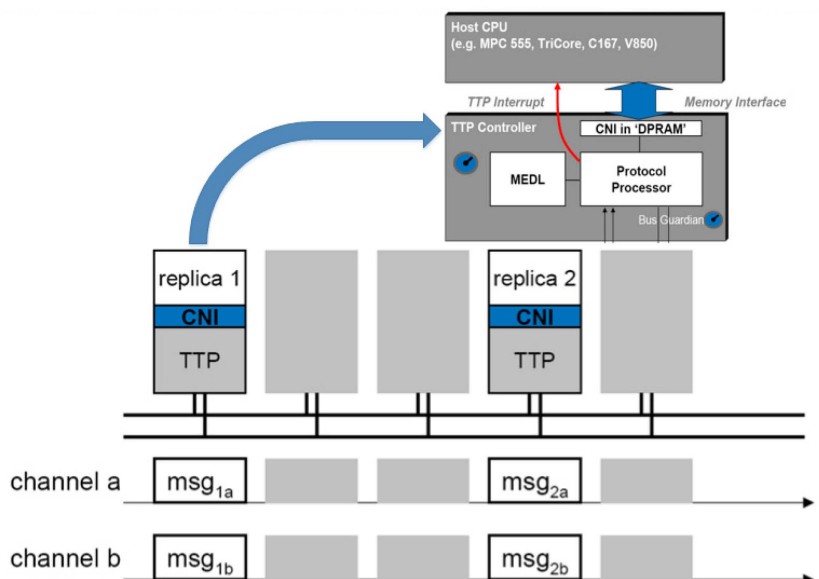

**Figure 3.** The architecture of TTP/C bus and Smart Node.

In this paper, the TTP/C bus network model and node data transceiver model are established by the SimEvents toolbox. Figure 4 shows the TTP controller model of the smart node. A TTP controller model is combined with a MEDL module, communication network interface (CNI) module, receiver module, transmitter module, and bus guardian (BG) module. There are two "entities" in this network model: one is the TTP bus data, and the other is the MEDL information. The MEDL decides the TTP controller which is the right time slot to broadcast data onto TTP bus, and CNI is responsible for the interaction with the host controller. The models of the engine, sensors, host control CPU of SNs, and actuators are established by the Simulink toolbox and the Stateflow toolbox. Finally, all related models are gathered and realized in the Simulink platform to build the integrated engine DCS model (shown in Figure 5), to meet the development and verification requirements of an aero-engine distributed control system. The integrated engine DCS model features dual channel system architecture, and each channel can individually control the engine with full authority and functionality. Each channel includes a central controller (EEC node), a front digital concentrator (DC1 node), a rear digital concentrator (DC2 node), sensors, and smart actuators. Pressure sensors are integrated in the digital concentrator. Other sensors such as LVDT, RVDT, thermal resistance, or thermocouple are traditional sensors, and their raw signals are collected by a digital concentrator to conduct AC/DC conversion and diagnosis, then packaged to broadcast to TTP network. The central controller is the brain of the whole system, which calculates the big closed-loop demands and makes decisions for the whole of the manipulations. The smart actuator node such as smart electric pump and RDDV valve receive the demands and control the fuel or location in small closed-loop.

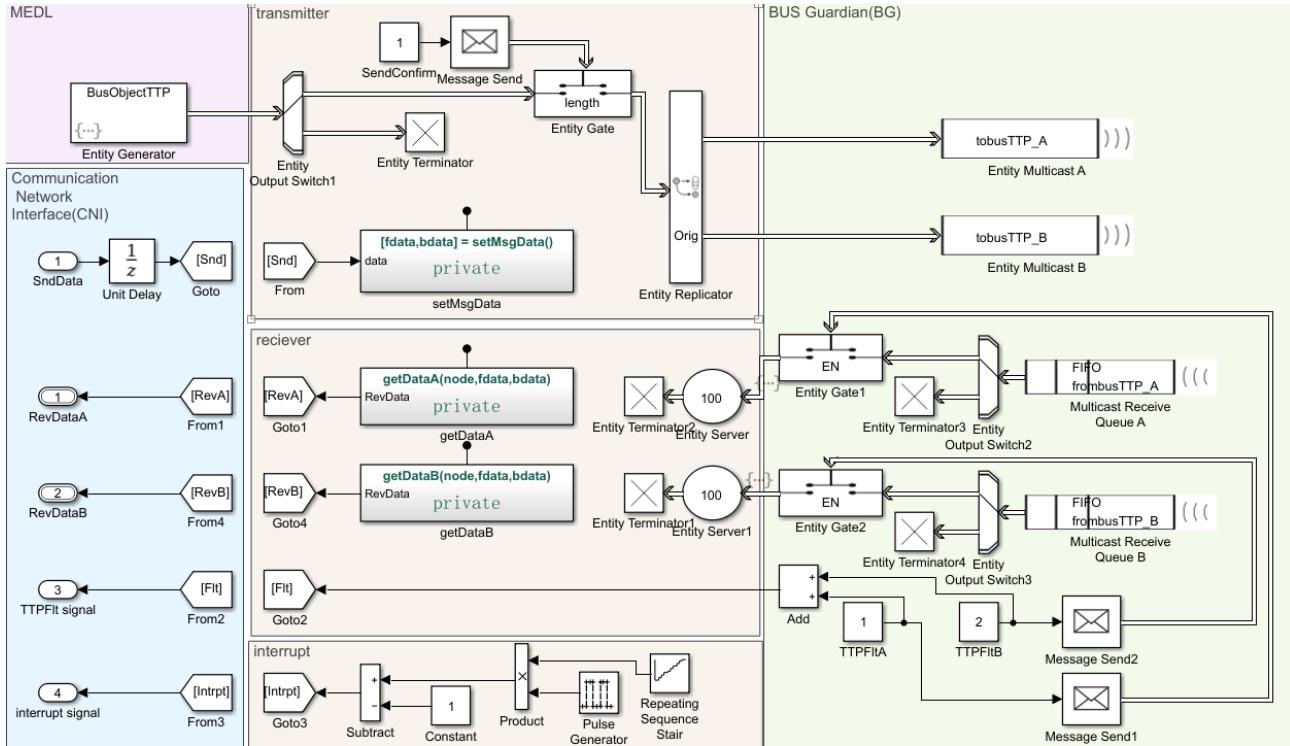

**Figure 4.** The smart node TTP controller model.

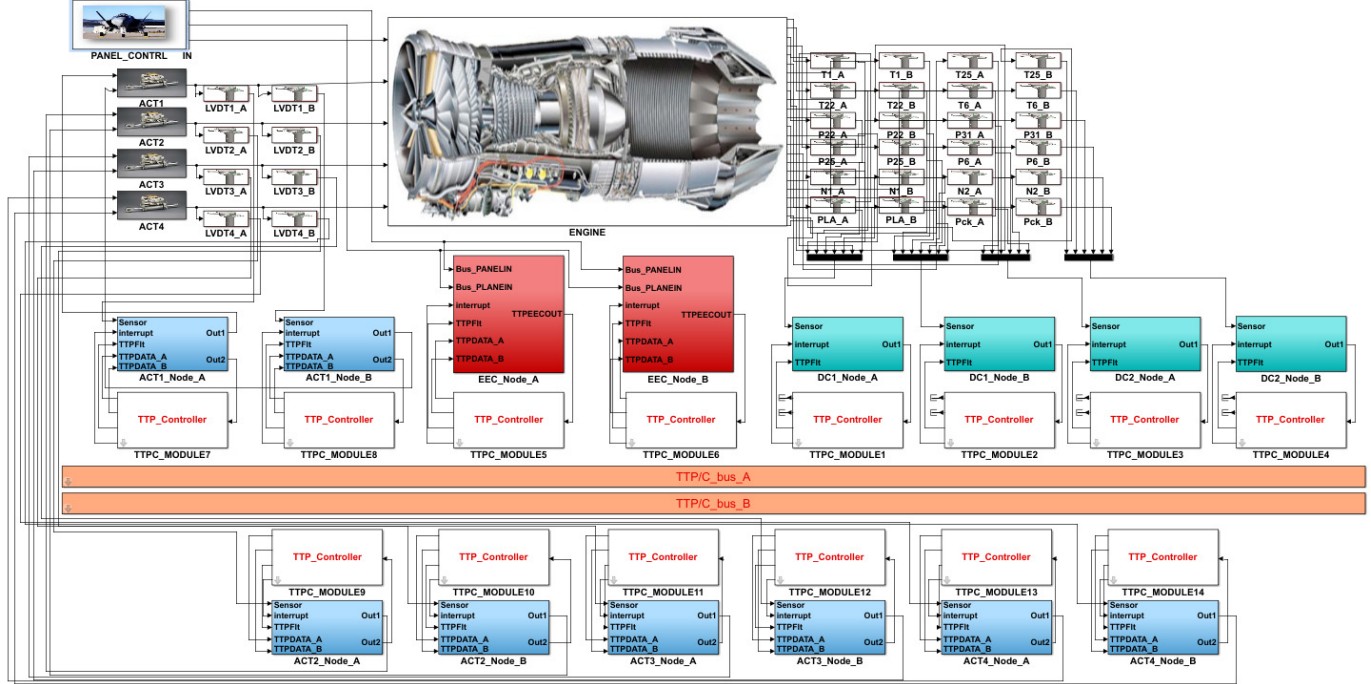

**Figure 5.** The numerical simulation environment of engine distributed control system.

## 4. Fault Diagnosis of Aero-Engine Distributed Control System Based on Sliding Mode Observer

Consider aero-engine state space model with health parameters and sensor faults:

$$\begin{aligned}
\dot{x}(t) &= Ax(t) + Bu(t) + Lh(t) \\
y(t) &= Cx(t) + Du(t) + Mh(t) + f(t)
\end{aligned} \tag{5}$$

where $L \in \mathbb{R}^{n \times q}$, and $M \in \mathbb{R}^{p \times q}$ ($q = 7$) are constant coefficient matrices, and $f$ denotes the sensor fault vector with dimension equal to $y$. Since $y = [N_L, N_H, P_{25}, T_{25}, P_3, T_3, T_{495}]^T$, $f = [f_1, f_2, f_3, f_4, f_5, f_6, f_7]^T$, where $f_i$ denotes the deviation fault of each sensor. Other definition and dimension of each appropriate dimension matrix and system variable corresponding to the Equation (5) have been given by the analysis in Section 2, where the health parameter and its derivative and the sensor fault and its derivative are unknown but bounded:

$$\|f(t)\| < \alpha_1, \|\dot{f}(t)\| < \beta_1, \|h(t)\| < \alpha_2, \|\dot{h}(t)\| < \beta_2 \tag{6}$$

where $\alpha_1$, $\alpha_2$, $\beta_1$, and $\beta_2$ are known real scalars. The notation $\|\cdot\|$ represents the Euclidean norm for vectors and the induced spectral norm for matrices.

In the existing literature, robust sliding mode reconstruction often requires the output dimension to be larger than the fault vector dimension. However, in the case discussed here, there are seven available sensors, and the dimension of $h$ and $f$ are both 7. Therefore, either in component fault or sensor fault cases, there is no such robust design freedom mentioned in the relevant literature. In view of this problem, a robust SMO design method for system with limited number of sensors using model construction and coordinate transformation is proposed in this chapter, so as to achieve robust fault reconstruction under any bounded uncertainty.

### 4.1. Introduction of Robust Design Degrees of Freedom under Limited Sensor Conditions

This section is concerned with the design of a SMO for an uncertain state variable model (SVM) of an aero-engine subject to component fault, and the result of which can be also applied to sensor fault diagnosis. A representative of the engine dynamic nonlinear model in a small range around steady-state operating point can be expressed as

$$\begin{aligned} \dot{x}(t) &= Ax(t) + Bu(t) + Lh(t) + Q_1\xi(t, x, u) \\ y(t) &= Cx(t) + Du(t) + Mh(t) + Q_2\xi(t, x, u) \end{aligned} \tag{7}$$

where $Q_1 \in \mathbb{R}^{n \times r}$ and $Q_2 \in \mathbb{R}^{p \times r}$ represent the uncertainty distribution matrix, while $\xi(t) \in \mathbb{R}^{r \times 1}$ denotes uncertainties. Assume $\xi(t)$ and its first-time derivatives are unknown but bounded

$$\|\xi(t, x, u)\| < \pi_1, \ \|\dot{\xi}(t, x, u)\| < \pi_2 \tag{8}$$

where $\pi_1$, $\pi_2$ are known real scalars. Before observer design, a linear transformation is introduced to $y(t)$ to create

$$y_V(t) = Vy(t) \tag{9}$$

where $V \in \mathbb{R}^{(p+1) \times p}$ is a designed matrix with a full column rank, and $y_V(t) \in \mathbb{R}^{(p+1) \times 1}$ is the augmented output. Since $V$ has a full column rank, the left pseudo-inverse of $V$ is well defined. Then $y(t)$ can be directly calculated as

$$y(t) = \left(V^T V\right)^{-1} V^T y_V(t) \tag{10}$$

This indicates a one-to-one correspondence between $y(t)$ and $y_V(t)$. In Equation (7), $y(t)$ is substituted by $y_V(t)$ to obtain the state-space equation of the following form

$$\begin{aligned} \dot{x}(t) &= Ax(t) + Bu(t) + Lh(t) + Q_1\xi(t, x, u) \\ y_V(t) &= VCx(t) + VDu(t) + VMh(t) + VQ_2\xi(t, x, u) \end{aligned} \tag{11}$$

In this way, $y_V(t)$ exceeds $h(t)$ in dimension, and $y_V(t)$ retains all measurement information in $y(t)$. Although $y_V(t)$ has no physical meaning in itself, it creates a degree of freedom in robust design structurally. Then $h(t)$ is regarded as the unknown input of the system, and the unknown input sliding mode observer can be constructed to reconstruct the fault in real time.

In order to make $h(t)$ and $\xi(t, x, u)$ appear only in the system equation, according to reference [26], a filter transformation and a coordinates change should be introduced to Equation (11) to get the following state-space equation:

$$\dot{x}_b(t) = A_b x_b(t) + B_b u(t) + H_b h(t) + Q_b \xi(t)$$
$$z_{at}(t) = C_b x_b(t) \tag{12}$$

Equation (12) is a canonical form from [26], which constitutes a useful starting point for observer design. The detail deduction can be found in the Appendix A.

### 4.2. Robust Second-Order SMO with Super-Twisting Algorithm

Define $e_z(t) = \hat{z}_{at}(t) - z_{at}(t)$ as output estimation error, where $\hat{z}_{at}(t)$ is the estimate value of $z_{at}(t)$. The proposed SMO has the following structure

$$\dot{\hat{x}}_b(t) = A_b \hat{x}_b(t) + B_b u(t) - G_l e_z(t) + G_n v(t)$$
$$\hat{z}_{at}(t) = C_b \hat{x}_b(t) \tag{13}$$

where $\hat{x}_b(t)$ is the estimate value of $x_b(t)$. $G_l \in \mathbb{R}^{\tilde{n} \times \tilde{p}}$, $G_n \in \mathbb{R}^{\tilde{n} \times \tilde{p}}$ are linear gain matrix and nonlinear gain matrix, respectively. The definition of $\tilde{n}$ and $\tilde{p}$ can be found in the Appendix A. Define $e_z(t) = \left[ e_{z,1}(t), e_{z,2}(t), .., e_{z,\tilde{p}}(t) \right]^{\mathrm{T}}$, then the discontinuous switching term $v(t) = \left[ v_1(t), v_2(t), .., v_{\tilde{p}}(t) \right]^{\mathrm{T}}$ is defined component-wise as

$$v_i(t) = -\psi_i sign(e_{zi}(t)) |e_{zi}(t)|^{1/2} + d_i(t)$$
$$\dot{d}_i(t) = -\varsigma_i sign(e_{zi}(t)) - \varphi_i e_{zi}(t) , \quad i = 1, 2, .., \tilde{p} \tag{14}$$

where $\psi$, $\varsigma$, and $\varphi$ are design scalars to be chosen. Assume that $G_n$ has the structure

$$G_n = \begin{bmatrix} -ET^{\mathrm{T}} \\ T^{\mathrm{T}} \end{bmatrix} \tag{15}$$

where $E \in \mathbb{R}^{(\tilde{n}-\tilde{p}) \times \tilde{p}}$ represents the design freedom. A special structure is imposed on $E$

$$E = [E_1, 0] \tag{16}$$

with $E_1 \in \mathbb{R}^{(\tilde{n}-\tilde{p}) \times (\tilde{p}-\tilde{q})}$. Obviously only when $\tilde{p} > \tilde{q}$ (sensor outnumbers fault) $E_1$ (robust design freedom) could exist. Define $e(t) = \hat{x}_b(t) - x_b(t)$ as state estimation error. The following error system is obtained from Equations (12) and (13)

$$\dot{e}(t) = A_b e(t) - G_l e_z(t) + G_n v(t) - H_b h(t) - Q_b \xi(t, x, u) \tag{17}$$

According to the form of $C_b$, $e(t)$ can be partition as $\left[ e_1^{\mathrm{T}}(t), e_2^{\mathrm{T}}(t) \right]^{\mathrm{T}}$ where $e_1(t) \in \mathbb{R}^{\tilde{n}-\tilde{p}}$. Let $G_l = \begin{bmatrix} G_{l1} \\ G_{l2} \end{bmatrix}$ where $G_{l1} \in \mathbb{R}^{(\tilde{n}-\tilde{p}) \times \tilde{p}}$, and $Q_b = \begin{bmatrix} Q_{b1} \\ Q_{b2} \end{bmatrix}$ where $Q_{b1} \in \mathbb{R}^{(\tilde{n}-\tilde{p}) \times \tilde{r}}$, then the error system can be written as

$$\begin{bmatrix} \dot{e}_1(t) \\ \dot{e}_2(t) \end{bmatrix} = \begin{bmatrix} A_{b11} & A_{b12} \\ A_{b21} & A_{b22} \end{bmatrix} \begin{bmatrix} e_1(t) \\ e_2(t) \end{bmatrix} - \begin{bmatrix} G_{l1} \\ G_{l2} \end{bmatrix} e_z(t) + \begin{bmatrix} -ET^{\mathrm{T}} \\ T^{\mathrm{T}} \end{bmatrix} v(t)$$
$$- \begin{bmatrix} 0 \\ H_{b2} \end{bmatrix} h(t) - \begin{bmatrix} Q_{b1} \\ Q_{b2} \end{bmatrix} \xi(t) \tag{18}$$

Consider a further coordinate transformation associated with the invertible matrix

$$T_L = \begin{bmatrix} I_{\tilde{n}-\tilde{p}} & E \\ 0 & T \end{bmatrix} \tag{19}$$

Then

$$\bar{e}(t) = T_L \begin{bmatrix} e_1(t) \\ e_2(t) \end{bmatrix} = \begin{bmatrix} e_1(t) + Ee_z(t) \\ e_z(t) \end{bmatrix} = \begin{bmatrix} \bar{e}_1(t) \\ e_z(t) \end{bmatrix} \tag{20}$$

Thus, the error system in Equation (17) can be written in the new coordinates as

$$\begin{bmatrix} \dot{\bar{e}}_1(t) \\ \dot{e}_z(t) \end{bmatrix} = \underbrace{\begin{bmatrix} \overline{A}_{b11} & \overline{A}_{b12} \\ \overline{A}_{b21} & \overline{A}_{b22} \end{bmatrix}}_{\overline{A}_b} \begin{bmatrix} \bar{e}_1(t) \\ \bar{e}_z(t) \end{bmatrix} - \underbrace{\begin{bmatrix} \overline{G}_{l1} \\ \overline{G}_{l2} \end{bmatrix}}_{\overline{G}_l} e_z(t) + \underbrace{\begin{bmatrix} \mathbf{0} \\ I_{\widetilde{p}} \end{bmatrix}}_{\overline{G}_n} \nu(t)$$
$$- \underbrace{\begin{bmatrix} \mathbf{0} \\ \overline{H}_{b2} \end{bmatrix}}_{\overline{H}_b} H(t) - \underbrace{\begin{bmatrix} \overline{Q}_{b1} \\ \overline{Q}_{b2} \end{bmatrix}}_{\overline{Q}_b} \xi(t, x, u) \tag{21}$$

where $\overline{A}_{b11} = A_{b11} + EA_{b21}$, $\overline{A}_{b21} = TA_{b21}$. Then a choice of the linear gain $\overline{G}_l$ is of the form

$$\overline{G}_l = \begin{bmatrix} \overline{G}_{l1} \\ \overline{G}_{l2} \end{bmatrix} = \begin{bmatrix} \overline{A}_{b12} \\ \overline{A}_{b22} + diag[\chi_1, \chi_2, .., \chi_{\widetilde{p}}] \end{bmatrix} \tag{22}$$

where $\chi$ is a scalar to be chosen. Substituting Equation (22) into Equation (20) yields

$$\begin{bmatrix} \dot{\bar{e}}_1(t) \\ \dot{e}_z(t) \end{bmatrix} = \begin{bmatrix} \overline{A}_{b11} & \mathbf{0} \\ \overline{A}_{b21} & -diag[\chi_1, \chi_2, .., \chi_{\widetilde{p}}] \end{bmatrix} \begin{bmatrix} \bar{e}_1(t) \\ \bar{e}_z(t) \end{bmatrix} + \begin{bmatrix} \mathbf{0} \\ I_{\widetilde{p}} \end{bmatrix} \nu(t)$$
$$- \begin{bmatrix} \mathbf{0} \\ \overline{H}_{b2} \end{bmatrix} h(t) - \begin{bmatrix} \overline{Q}_{b1} \\ \overline{Q}_{b2} \end{bmatrix} \xi(t, x, u) \tag{23}$$

provided the structure of $E$ in Equation (15), $\overline{A}_{b11}$ can be written as $A_{b11} + E_1 A_{b211}$, where $A_{b211}$ is the first $\widetilde{p} - \widetilde{q}$ row of $A_{b21}$. As argued in [27], if condition (2) is satisfied, then the pair $(A_{b11}, A_{b211})$ is detectable. Suppose that $E$ in accord with Equation (16) has been chosen such that $\overline{A}_{b11}$ is stable, i.e., there exists a symmetric positive definite matrix $P_{11} \in \mathbb{R}^{(\widetilde{n}-\widetilde{p}) \times (\widetilde{n}-\widetilde{p})}$ such that

$$\overline{A}_{b11}^T P_{11} + P_{11} \overline{A}_{b11} < 0 \tag{24}$$

The objective is to force the output error $e_z(t)$ to zero in finite time and induce a sliding mode on the sliding manifold

$$S = \left\{ \begin{bmatrix} \bar{e}_1^T(t) & e_z^T(t) \end{bmatrix}^T | e_z(t) = 0 \right\} \tag{25}$$

Considering the structure of $\nu(t)$ in Equation (14), and substituting Equation (14) into Equation (23), the equation related to $e_z(t)$ in Equation (23) can be written component-wise as

$$\dot{e}_{zi}(t) = -\psi_i sign(e_{zi}(t))|e_{zi}(t)|^{1/2} - \chi_i e_{zi}(t) + \overline{A}_{b21i}\bar{e}_1(t)$$
$$- \overline{H}_{b2i}h(t) - \overline{Q}_{b2i}\xi(t, x, u) + d_i(t) \tag{26}$$

$$\dot{d}_i(t) = -\varsigma_i sign(e_{zi}(t)) - \varphi_i e_{zi}(t) , \quad i = 1, 2, .., \widetilde{p}$$

where $\overline{A}_{b21,i}$, $\overline{H}_{b2,i}$, and $\overline{Q}_{b2,i}$ are the $i^{th}$ row of $\overline{A}_{b21}$, $\overline{H}_{b2}$, and $\overline{Q}_{b2}$, respectively. By defining a new variable

$$d_{0i}(t) = \overline{A}_{b21i}\bar{e}_1(t) - \overline{H}_{b2i}h(t) - \overline{Q}_{b2i}\xi(t, x, u) + d_i(t), \quad i = 1, 2, .., \widetilde{p} \tag{27}$$

the Equation (26) can be rewritten as

$$\dot{e}_{zi}(t) = -\psi_i sign(e_{zi}(t))|e_{zi}(t)|^{1/2} - \chi_i e_{zi}(t) + d_{0i}(t)$$
$$\dot{d}_{0i}(t) = -\varsigma_i sign(e_{zi}(t)) - \varphi_i e_{zi}(t) + \phi_i(t) , \quad i = 1, 2, .., \widetilde{p} \tag{28}$$

where $\phi_i(t) = \overline{A}_{b21,i}\dot{\overline{e}}_1(t) - \overline{H}_{b2,i}\dot{h}(t) - \overline{Q}_{b2,i}\dot{\xi}(t)$. Then

$$\|\phi_i(t)\| < \|\overline{A}_{b21i}\| \cdot \|\dot{\overline{e}}_1(t)\| + \|\overline{H}_{b2i}\| \cdot \|\dot{h}(t)\| + \|\overline{Q}_{b2i}\| \cdot \|\dot{\xi}(t,x,u)\| \tag{29}$$

Since $\overline{A}_{b11}$ is stable by assumption in Equation (24), the autonomous system associated with $\overline{e}_1(t)$ is stable. Consequently both $\|\overline{e}_1(t)\|$ and $\|\dot{\overline{e}}_1(t)\|$ are bounded. Provided $\|\dot{h}_1(t)\|$ and $\|\dot{\xi}(t)\|$ are bounded, then there exists a sufficiently large $\varepsilon$ with which $\|\phi_i(t)\| < \varepsilon$ is satisfied. As discussed in [28], choose the scalar gains from Equation (26) as

$$\psi_i > 2\sqrt{\varepsilon_i}, \chi_i > 0, \varsigma_i > \varepsilon_i, \varphi > \frac{\chi^2(\psi_i^3 + 5/4\psi_i^2 + 5/2(\varsigma_i - \varepsilon_i))}{\psi_i(\varsigma_i - \varepsilon_i)} \tag{30}$$

It can be proven that a sliding motion will take place and $\dot{e}_{z,i}(t) = e_{z,i}(t) = 0$ in finite time.

### 4.3. The Design of Variable Gain Discontinuous Switching Term

From the form of $v(t)$ it is obvious that $\psi$ and $\varsigma$ is the nonlinear part gains and they determine the system chattering level, and the values of $\psi$ and $\varsigma$ are all related with $\varepsilon$. Therefore, in order to reduce chattering, the variable gain discontinuous switching term is put forward here, the improved form of observer gain is as

$$\psi_i(t) = 2\delta\sqrt{\hat{\varepsilon}_i(t)}, \ \varsigma_i(t) = \delta\hat{\varepsilon}_i(t) \tag{31}$$

where $\delta > 1$ is the safety coefficient, and $\hat{\varepsilon}_i$ is the estimate value of $\varepsilon_i$, which is

$$\hat{\varepsilon}_i(t) = \|\overline{A}_{b21i}\| \cdot \|\dot{\overline{e}}_1(t)\| + \|\overline{H}_{b2i}\| \cdot \|\dot{\hat{h}}(t)\| + \|\overline{Q}_{b2i}\| \cdot \pi_2 \tag{32}$$

In this way, the value of the discontinuous switching term varies depending on $\hat{\varepsilon}_i$. In cases when there is no fault or minor fault, chattering could be restricted to a reasonable level. Meanwhile, if disturbances or fault are severe, the adaptive gain is able to guarantee the stability of sliding modes.

### 4.4. The Solve of Observer Gain by LMI

Once the sliding surface is reached, the error dynamics in Equation (23) are given by

$$\begin{aligned} \dot{\overline{e}}_1(t) &= \overline{A}_{b11}\overline{e}_1(t) - \overline{Q}_{b1}\xi(t) \\ 0 &= \overline{A}_{b21}\overline{e}_1(t) + I_{\tilde{p}}v_{eq}(t) - \overline{H}_{b2}h(t) - \overline{Q}_{b2}\xi(t) \end{aligned} \tag{33}$$

Provided $\overline{Q}_{b1} = Q_{b1} + EQ_{b2}$, $\overline{Q}_{b2} = TQ_{b2}$, and $\overline{H}_{b2} = TH_{b2}$, the Equation (33) can be rearranged and rewritten as

$$\begin{aligned} \dot{\overline{e}}_1(t) &= (A_{b11} + EA_{b21})\overline{e}_1(t) - (Q_{b1} + EQ_{b2})\xi(t) \\ v_{eq}(t) &= -TA_{b21}\overline{e}_1(t) + TH_{b2}h(t) + TQ_{b2}\xi(t) \end{aligned} \tag{34}$$

where the signal $v_{eq}(t)$ is the so-called equivalent output injection signal. As in [27], $v_{eq}(t)$ represents the averaged behavior of $v(t)$ and is required to maintain a sliding motion, which can be obtained by

$$\dot{v}_{eq}(t) = -A_{fv}v_{eq}(t) + A_{fv}v(t) \tag{35}$$

where $A_{fv}$ is the filter matrix and $A_{fv}$ is in the form of a diagonal matrix with positive entries where the diagonal elements represent inverse time constants.

Define a weighting matrix $W$ in the structure of

$$W = \begin{bmatrix} W_1 & H_{b0}^{-1} \end{bmatrix} \tag{36}$$

where $W_1 \in \mathbb{R}^{\tilde{q} \times (\tilde{p}-\tilde{q})}$ represents design freedom. Then an estimation signal is defined as

$$\hat{h}(t) = WT^{\mathrm{T}} v_{eq}(t) \tag{37}$$

Note that $WH_{b2} = I_{\tilde{p}}$. Multiplying the second equation in Equation (34) with $WT^{\mathrm{T}}$ and rearranging Equation (34) yields

$$\begin{aligned} \dot{\overline{e}}_1(t) &= (A_{b11} + EA_{b21})\overline{e}_1(t) - (Q_{b1} + EQ_{b2})\boldsymbol{\xi}(t) \\ \hat{h}(t) - h(t) &= -WA_{b21}\overline{e}_1(t) + WQ_{b2}\boldsymbol{\xi}(t,x,u) \end{aligned} \tag{38}$$

From Equation (38) it is clear that the transfer function from $\boldsymbol{\xi}(t,x,u)$ to $\hat{h}(t) - h(t)$ is

$$G(s) = WA_{b21}(sI_n - (A_{b11} + EA_{b21})^{-1})(Q_{b1} + EQ_{b2}) + WQ_{b2} \tag{39}$$

The objective is to minimize the effect of $\boldsymbol{\xi}(t,x,u)$ on the estimation error $\hat{h}(t) - h(t)$. In addition, note that the sliding surface can be reached only if Equation (24) is satisfied. Thus, the design is aimed at stabilizing $A_{b11} + EA_{b21}$ while minimizing the effect of $\boldsymbol{\xi}(t,x,u)$ on $\hat{h}(t) - h(t)$. Using the Bounded Real Lemma in [29], if there exists a matrix $P_{11}$ as defined in Equation (24), and another matrix $P_{12} \in \mathbb{R}^{(\tilde{n}-\tilde{p}) \times \tilde{p}}$ in the form of $\begin{bmatrix} P_{121} & 0 \end{bmatrix}$, where $P_{121} \in \mathbb{R}^{(\tilde{n}-\tilde{p}) \times (\tilde{p}-\tilde{q})}$, such that

$$\begin{bmatrix} P_{11}A_{b11} + A_{b11}^{\mathrm{T}}P_{11} + P_{12}A_{b21} + A_{b21}^{\mathrm{T}}P_{12}^{\mathrm{T}} & * & * \\ -(P_{11}Q_{b1} + P_{12}Q_{b2}) & -\gamma I_{\tilde{r}} & * \\ -WA_{b21} & WQ_2 & -\gamma I_{\tilde{q}} \end{bmatrix} < 0 \tag{40}$$

then $\left\| \hat{h}(t) - h(t) \right\| < \gamma \alpha_1$. It is a standard LMI problem which can be solved by function "mincx" in LMI toolbox [29]. Once $P_{11}$, $P_{12}$ is synthesized, $E$ is chosen as $P_{11}^{-1}P_{121}$, and it is obvious Equation (24) is satisfied. Then $G_n$ is obtained and $G_l$ can be calculated as $T_l^{-1}\overline{G}_l$. Although the influence of system uncertainty on the reconstructed signal cannot be completely decoupled, its transfer function is minimized through LMI optimization. In addition, if the output vector dimension is equal to the fault vector dimension, then there would be no degree of freedom for robust design such as $E_1$ and $W_1$. Therefore, it is necessary to increase the output dimension by linear transformation as depicted in 4.1.

Finally, the estimation of health parameter (or sensor fault) is given by the signal $\hat{h}(t)$ defined in Equation (37).

## 5. Simulations

The gain of the observer is calculated at the cruise design point, then digital simulations are carried out on the component-level model (CLM) and integrated model of engine DCS. The simulating step of the engine CLM is 5ms, while that of the closed-loop control is 20ms. The relevant parameters of the sliding mode observer are selected as follows: $A_f = 0.1I_7$, $A_{fv} = 4.5I_7$, $\chi_i = 0.005$, $\varphi_i = 0.1$ ($i = 1,2,..,7$). The variable gain method described in Section 4.3 is adopted for the selection of $\psi_i$ and $\varsigma_i$, and the safety factor $\delta = 1.5$.

Aero-engine gas path sensors are usually installed in high-temperature and high-pressure environments, plus, system vibration brought by high-speed rotating components are inevitable, and they are easily affected by electromagnetic interference and system noise. As a result, signals from sensors contain inherent noise, system noise and intrusion noise, among which system noise caused by engine vibration is the main one. Due to the symmetry of rotating parts, system noise is generally in the form of Gaussian white noise;

therefore, Gaussian white noise is adopted to simulate the measurement noise. Referring to the indicators of engine measurement noise in the existing literature [30,31], the magnitude of injected measurement noise in this paper is shown in Table 2.

**Table 2.** The magnitude of injected measurement noise.

| Sensor Type | Standard Deviation (SD)% |
| --- | --- |
| N1 | 0.15 |
| N2 | 0.20 |
| temperature sensor (<600 K) | 0.15 |
| temperature sensor (>600 K) | 0.20 |
| pressure sensor | 0.20 |

Two gas-path performance (health parameter) fault scenarios and two sensor fault scenarios are simulated, respectively. The proposed SMO is used to observe the current control system in real time, and results are shown in Figure 6. Scenario 1 simulates compressor fault, where flow decreases by 2% and efficiency decreases by 4.5%. Scenario 2 simulates high-pressure turbine fault, corresponding to a 2% increase in flow and a 2% decrease in efficiency. Scenario 3 simulates a −10% bias fault of T3 sensor; Scenario 4 simulates both N2 sensor and T495 sensor with −11% and −9% bias faults, respectively. In order to verify the estimation accuracy and the estimating speed of the observer, step faults are injected in the simulation to simulate abrupt engine faults. As can be seen from Figure 6, the sliding mode observer designed in this paper has achieved good fault estimation results in all fault scenarios. Table 3 shows the quantitative comparison between the proposed method and linear Kalman filter in RMSE, SD and estimated time. It can be seen that RMSE and SD of the two methods are basically at the same level. In terms of estimation time, the sliding mode observer based on the unknown input principle avoids the influence of fault derivative on observer convergence, and the estimation time is greatly shortened.

In dealing with sliding mode chattering, this paper optimized the observer design successively by super-twisting algorithm and adaptive gain method. Considering the fact that the low-pass filter featured with filter matrix $A_{fv}$ is adopted to recover $v_{eq}$ from $v$, and the low-pass filter can also inhibit high-frequency noise. Therefore, in order to independently verify the chattering suppression effect, measurement noise is not added in the simulation here, and other simulation conditions are the same as above. Figure 7 shows the lateral comparison using different observers when the fan efficiency decreases by 6%. Figure 7a shows the STA method without adaptive gain, where $\psi_i = 0.283$, $\varsigma_i = 0.02$ ($i = 1, 2, 7$), and other design parameters remain same with observers above. In Figure 7b, STA method with adaptive gain is adopted. It can be seen that some components of $\hat{h}$ have large chattering in Figure 7a, such as $SE4$ ($\hat{h}_7$). This is because $\psi_i$ or $\varsigma_i$ have the same value in each dimension, and the conservative design leads to the problem that the fixed values of $\psi_i$ or $\varsigma_i$ are too large for no-fault cases, which creates a large $v$ that causes chattering. The average SD of each dimension of $\hat{h}$ is $5.13 \times 10^{-4}$, and $\hat{h}_7$ is $9.33 \times 10^{-4}$. The chattering observed in Figure 7b is controlled within an ideal range, and the average SD of each dimension of $\hat{h}$ is $3.8 \times 10^{-5}$. At the same time, since the values of $\psi_i$ and $\varsigma_i$ will adaptively increase in the process of dynamic convergence, Figure 7b is significantly better than Figure 7a in the dynamic process of sliding mode convergence.

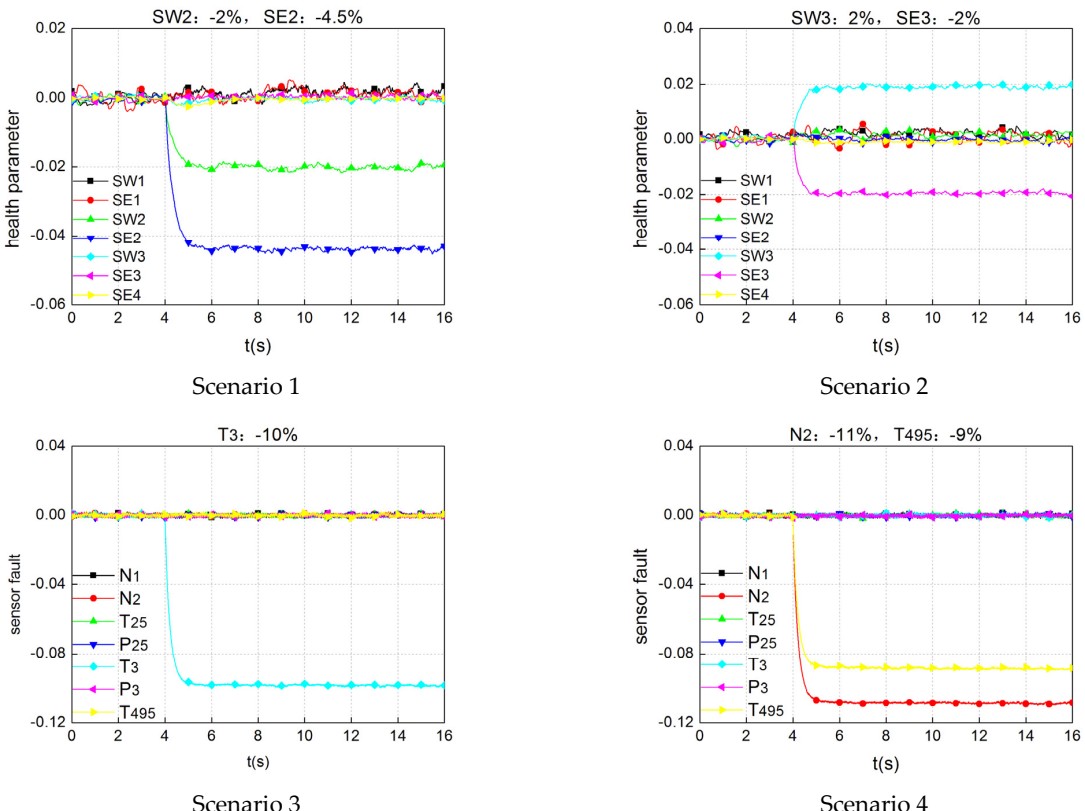

**Figure 6.** The simulation results of SMO.

**Table 3.** Quantitative comparison of SMO and KF.

| Scenario | RMSE ($10^{-4}$) | | SD ($10^{-4}$) | | Time (s) | |
|---|---|---|---|---|---|---|
| | SMO | LKF | SMO | LKF | SMO | LKF |
| 1 | 12.2 | 19.7 | 5.4 | 5.6 | 1.0 | 2.7 |
| 2 | 12.1 | 11.7 | 6.4 | 5.3 | 0.9 | 2.4 |
| 3 | 3.8 | 5.2 | 3.2 | 5.0 | 1.0 | 1.9 |
| 4 | 5.7 | 4.6 | 5.2 | 4.6 | 1.0 | 2.6 |

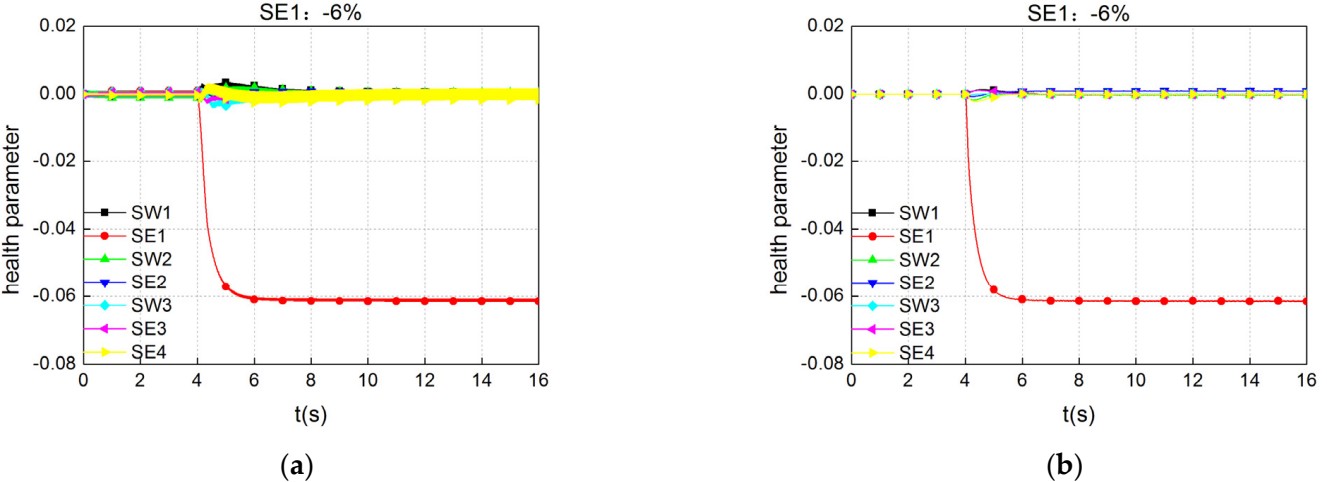

**Figure 7.** The simulation results of SMOs in view of chattering. (**a**) shows the STA method without adaptive gain; (**b**) shows the STA method with adaptive gain.

In order to verify the influence of the related uncertainties introduced by the distributed system (time delay, packet loss, disturbance) on the diagnosis and the robustness of the designed observer, the following form of uncertainty is added to the engine baseline model: $\boldsymbol{\xi} = \begin{bmatrix} \boldsymbol{\xi}_1 & \boldsymbol{\xi}_2 & \boldsymbol{\xi}_3 \end{bmatrix}^{\mathrm{T}}$, where $\boldsymbol{\xi}_1 \in R^{1 \times 1}$ and $\boldsymbol{\xi}_2 \in R^{1 \times 1}$ represent model deviation, and $\boldsymbol{\xi}_3 \in R^{1 \times 1}$ represents disturbance. Plus, the bus packaging model is used to simulate the bus time delay and random packet loss behavior, in which the simulated delay is 50 ms and the simulated packet loss rate is 1%. Let each element of $\boldsymbol{\xi}$ be:

$$\xi_1(t) = \begin{bmatrix} 0.21 & 0.11 \end{bmatrix} x_a(t), \ \xi_2(t) = \begin{bmatrix} 0.13 & -0.17 \end{bmatrix} x_a(t)$$
$$\xi_3(t) = 0.02 + 0.01\sin(\tfrac{\pi t}{3}) \tag{41}$$

and uncertainty matrices $Q_1$ and $Q_2$ are chosen as:

$$Q_1 = \begin{bmatrix} 1 & 0 & 0 \\ 0 & 1 & 0 \end{bmatrix}, Q_2 = \begin{bmatrix} 0 & 0 & 1 \\ 0 & 0 & 1 \\ & \boldsymbol{0} & \end{bmatrix} \tag{42}$$

The forms of $Q_1$ and $Q_2$ reflect that $\xi_1$ and $\xi_2$ represent the deviation of the system matrix *A*, while $\xi_3$ represents the interference in the form of sine function in the speed output channel. Figure 8 simulates fan flow (SW1) decreased by 5% and Figure 9 simulates T25 decreased by 3%. In order to facilitate comparison, the estimation results of SMO under this uncertainty, the results of SMO under no uncertainty and the simulation results of LKF under the same uncertainty are both given. It can be seen that the addition of uncertainty has almost no impact on the SD and estimated time of SMO, while the accuracy is slightly decreased. However, compared with LKF under the same simulation conditions, the accuracy of SMO is significantly higher, especially in the simulation shown in Figure 8. This is because the LKF method is only robust against Gaussian white noise with averaged value 0, and obviously in this case, the injected uncertainties are not Gaussian white noise. However, the proposed method shows strong robustness in view of bounded uncertainties, due to the transfer function of uncertainty onto the reconstructed signal is minimized through LMI optimization.

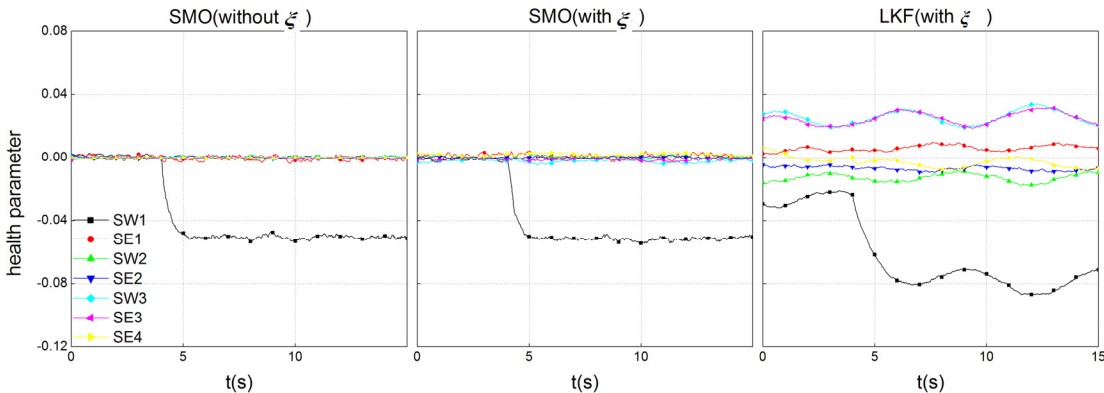

**Figure 8.** The comparison results under uncertainty (fan flow decreased by 5%).

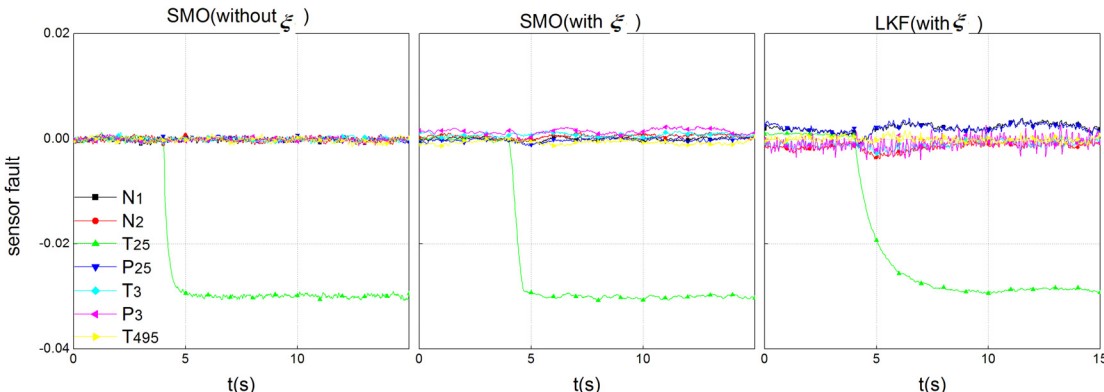

**Figure 9.** The comparison results under uncertainty (T25 decreased by 3%).

## 6. Conclusions

In this paper, a TTP/C bus packaging model supporting time-delay and packet loss simulation and timing planning is developed based on SimEvents. Combined with the development of component-level model, sensor, controller, and actuator model by Simulink, the proposed engine distributed control model is capable of verifying aero-engine DCS diagnosis method. Moreover, a robust gas-path diagnosis method for engine distributed systems is proposed using sliding mode observer, aiming at gas-path parameter estimation and fault diagnosis regardless of system uncertainties such as time-delay and packet loss. With health parameters or sensor faults modeled as artificial inputs, the described approach is applicable to both slow degradations and abrupt shifts. Considering the fact that the involved engine contains an equal number of available sensors and health parameters, a transformation has been introduced to create a fictitious output that dimensionally outnumbers the health parameter (or sensor fault) vector, which makes room for the robust design. Combining high-order sliding mode theory, observation matrix LMI optimization, and variable gain methods, chattering is suppressed and robust observation accuracy is improved. By comparing with the traditional Kalman filter, the simulation results show the advantages of the proposed method in distributed system application scenarios, and a range of fault cases can be faithfully detected and estimated, with suitable accuracy and quick diagnosis speed by the described method. The limitation of the proposed method in engine DCS application is that the considered plant is a commercial aero-engine, which is usually operated in cruise flight condition, and only steady-state of several typical state points are considered to be designed and verified. Further work can be expanded to jet engines and diagnosis in dynamic or a wider flight envelope should be researched.

**Author Contributions:** Conceptualization, X.C. and X.Q.; methodology, X.C. and X.Q.; software, X.C.; validation, X.C.; formal analysis, X.C.; investigation, X.Q.; resources, X.Q.; data curation, X.C.; writing—original draft preparation, X.C.; writing—review and editing, X.Q.; visualization, X.C.; supervision, X.Q.; project administration, X.Q.; funding acquisition, X.Q. All authors have read and agreed to the published version of the manuscript.

**Funding:** This research was funded by "Science Center for Gas Turbine Project, grant number P2022-A-V-001-001", and "National Major Project of Aero Engine and Gas Turbine, grant number J2019-V-0003-0094".

**Institutional Review Board Statement:** Not applicable.

**Informed Consent Statement:** Not applicable.

**Data Availability Statement:** Not applicable.

**Conflicts of Interest:** The authors declare no conflict of interest. The funders had no role in the design of the study; in the collection, analyses, or interpretation of data; in the writing of the manuscript; or in the decision to publish the results.

## Appendix A

In Section 4.1, a filter transformation and a coordinates change should be introduced to Equation (11), to make the further deduction on a nominal base form described in [26].

In order to make $h(t)$ and $\xi(t, x, u)$ appear only in the system equation, a filter transformation is introduced to $\mathbf{y}_V(t)$:

$$\dot{z}(t) = -A_f z(t) + A_f \mathbf{y}_V(t) \tag{A1}$$

where $-A_f \in \mathbb{R}^{(p+1) \times (p+1)}$ is a stable filter matrix. Typically, $A_f$ is in the form of a diagonal matrix with positive entries where the diagonal elements represent inverse time constants. Substituting $z_V(t)$ for $\mathbf{y}_V(t)$ in Equation (5), and combining $x(t)$ and $z_V(t)$ to create an augmented state $x_a(t) \in \mathbb{R}^{(n+p+1) \times 1}$, the following representation can be obtained

$$
\begin{bmatrix} \dot{x} \\ \dot{z}_V \end{bmatrix} = \underbrace{\begin{bmatrix} A & 0 \\ A_f VC & -A_f \end{bmatrix}}_{A_a} \underbrace{\begin{bmatrix} x \\ z_V \end{bmatrix}}_{x_a} + \underbrace{\begin{bmatrix} B \\ A_f VD \end{bmatrix}}_{B_a} u + \underbrace{\begin{bmatrix} L \\ A_f VM \end{bmatrix}}_{H_a} h + \underbrace{\begin{bmatrix} Q_1 \\ A_f VQ_2 \end{bmatrix}}_{Q_a} \xi
$$

$$
z_{at} = \underbrace{\begin{bmatrix} \tau_1 I_n & \tau_2 I_n & 0 \\ 0 & 0 & I_{p+1-n} \end{bmatrix}}_{C_{at}} \begin{bmatrix} x \\ z_V \end{bmatrix}
\tag{A2}
$$

where $A_a \in \mathbb{R}^{\tilde{n} \times \tilde{n}}$, $B_a \in \mathbb{R}^{\tilde{n} \times \tilde{m}}$, $H_a \in \mathbb{R}^{\tilde{n} \times \tilde{q}}$, $Q_a \in \mathbb{R}^{\tilde{n} \times \tilde{r}}$, and $C_a \in \mathbb{R}^{\tilde{p} \times \tilde{n}}$ are coefficient matrices, and $I_{\tilde{p}} \in \mathbb{R}^{\tilde{p} \times \tilde{p}}$ denotes identity matrix. Comparing to the original system in Equation (7), it gives $\tilde{n} = n + p + 1$, $\tilde{m} = m$, $\tilde{q} = q$, $\tilde{r} = r$, and $\tilde{p} = p + 1$. As analyzed in [16], it can be proved that the necessary and sufficient conditions for the existence of a stable sliding motion and feasibility of fault reconstruction [26] can be satisfied by Equation (A2). Thus, there exists an invertible change of coordinates $x_b(t) = T_b x_a(t)$, in which $C_{at}$ and $H_a$ have transformed to the following structure.

$$
C_b = C_{at} T_b^{-1} = \begin{bmatrix} 0 & T \end{bmatrix}, \quad H_b = T_b H_a = \begin{bmatrix} 0_{(\tilde{n}-\tilde{p}) \times \tilde{q}} \\ H_{b2} \end{bmatrix} = \begin{bmatrix} 0_{(\tilde{n}-\tilde{p}) \times \tilde{q}} \\ 0_{(\tilde{p}-\tilde{q}) \times \tilde{q}} \\ H_{b0} \end{bmatrix}
\tag{A3}
$$

where $T \in \mathbb{R}^{\tilde{p} \times \tilde{p}}$ is orthogonal, $H_{b0} \in \mathbb{R}^{\tilde{q} \times \tilde{q}}$ is non-singular, and $H_{b2} \in \mathbb{R}^{\tilde{p} \times \tilde{q}}$. With the change of coordinates the Equation (A2) is given by

$$
\begin{aligned}
\dot{x}_b(t) &= A_b x_b(t) + B_b u(t) + H_b h(t) + Q_b \xi(t) \\
z_{at}(t) &= C_b x_b(t)
\end{aligned}
\tag{A4}
$$

where $A_b = T_b A_a T_b^{-1}$ and $B_b = T_b B_a$. $A_b$ is in the form of $\begin{bmatrix} A_{b11} & A_{b12} \\ A_{b21} & A_{b22} \end{bmatrix}$ where $A_{b11} \in \mathbb{R}^{(\tilde{n}-\tilde{p}) \times (\tilde{n}-\tilde{p})}$. Equation (A4) is a canonical form from [26], which constitutes a useful starting point for observer design.

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
