# Peer review of "Robust Gas-Path Fault Diagnosis with Sliding Mode Applied in Aero-Engine Distributed Control System"

_sustainability, doi:10.3390/su151310278_

Round 1

Reviewer 1 Report

In this study, software that provides error analysis and control for aero engine has been developed and its results are given by four scenario.

First of all, it is useful to give the numerical data about the study in the abstract. The literature search can be expanded. In the literature section, information about what kind of errors may occur for the aero engine and about the sensors and software used can be added.

An abbreviation list can be created for the abbreviations in Figure 2 and all abbreviations in the manuscript. The word in this paper is used too much, it should be deleted from unnecessary places.

Figures 4 and 5 should be explained in great detail. The types of sensors used and which sensor is located where in the system.

What is the type of gas path detectors? The measuring range can be given in terms of precision. The functions of the sensors given in Figure 6 should be explained in detail. Values can be given in terms of their measurement sensitivities, uncertainty analysis and error analysis. Figures 8 and 9 should be explained in more detail. The results obtained should be explained and the results obtained should be interpreted. In addition, more detailed information can be given about the work that can be done in the future.

Reviewer 2 Report

The topic of the paper is great and I endorse the authors for such contribution. However, the content of the paper requires some manipulation before acceptance. So, the following comments have to be revised before the acceptance:

  1. The major drawback of the paper is its literature review. There are old and few research that referenced in the introduction section as a brief review over the related research. Please augment the review of related research section and for further detail, read the following papers and cite them suitably in your research:
    1. Zarandi, M. F., Soltanzadeh, S., Mohammadi, A., & Castillo, O. (2019). Designing a general type-2 fuzzy expert system for diagnosis of depression. Applied Soft Computing, 80, 329-341.
    2. Delaram, J., Houshamand, M., Ashtiani, F., & Valilai, O. F. (2021). A utility-based matching mechanism for stable and optimal resource allocation in cloud manufacturing platforms using deferred acceptance algorithm. Journal of Manufacturing Systems60, 569-584.
    3. Lv, C., Chang, J., Bao, W., & Yu, D. (2022). Recent research progress on airbreathing aero-engine control algorithm. Propulsion and Power Research.
  2. Figures 4 and 5 are unreadable. Please enhance the quality of the figures.
  3. The conclusion is too short and did not refer to the main findings of the paper. Furthermore, the conclusion is very similar to the abstract! You should rewrite the section. Please improve the structure of the conclusion section. Also, you should provide some future direction at the end of the paper for the interested audiences.
  4. In line 325, the paper stated “?1 ?1” is it a product of two parameters or they are separated? There are too many typos throughout the paper. Please conduct a complete proof-read throughout the paper.
  5. Equation 33 should be revised based on the Equation 29.

The writing of the paper should be revised meticulously. 

Reviewer 3 Report

This article initially develops a distributed bus packaging model that supports time-delay and packet-loss simulating and timing planning based on SimEvent, and the Sliding Mode Observer (SMO) is adopted to design a robust gas-path diagnosis method. This article has been well organized, but the following issues should be considered before the publication.

1. Page 3, lines 102-103. What is the specific meaning of the listed variables such as NL, NH and P25? The authors should add further explanation. 

2. Are the obtained health parameters related to the given measurements (such as NL, NH and P25)? The authors should be more specific in explaining how the calculation of the health parameters, such as SE and SW. 

3. The resolution of Figure 4 and Figure 5 is too low. I suggest the authors replace them. 

4. The proposed method is only compared with the linear Kalman filter (LKF). The effectiveness and superiority of the proposed method are not convincing enough. I suggest the authors supplement more comparison results, including some more advanced observer design methods

5. What are the limitations of the proposed method? The authors should discuss the potential shortcomings of the proposed method. 

Reviewer 4 Report

The authors describe a distributed control system for aeroengine with component-based modelling using a robust sliding mode observer method. The article presents a case study showing an improvement compared to what is stated to be a more common approach based on linear Kalman filter.

The impact of the study in terms of applicability of the results and value of the research is unclear, since the analysis is rather specific, and the engine model is quite lean.The research scope is limited and the novelty of the method is modest. This makes envisaging suggestions to improve the scholarship contribution of the paper quite difficult.

Nonetheless, it might be of interest to the industrial community to improve the actual level of control systems design.

The following suggestions are given in case the editor grants a revision:

The paper is sometimes hard to read, especially in the sections related with methods, because of the abuse of acronyms and a rather dense mathematical description. Minor language errors (spelling, syntax) are present. Please also address the following items:

11. Expand acronyms the first time they are employed, e.g., line 69 LVP and 81 LKF.

2. Line 103, the symbols are not defined.

3. Line 122, what parts are the indexes of h referring to?

4. The article is overly abundant of acronyms, which makes reading rather difficult. Try to avoid recurring to them when possible and add a nomenclature.

5. Page 5, figures 4 and 5 appear at low res and are hardly readable. Please ensure that a high-res versions is adopted in a final version of the manuscript.

6. What is Truetime toolbox at line172? Toolboxes are referred several times in that paragraph. Specify the software they are part of and provide appropriate citation, e.g., Simulink at line 176 or any commercial package employed.

7. Fault vector f in eq. 5, how is it defined?

8. Section 4 is dense of mathematics. Consider condensing only the final results of each subsection and letting the full derivation of the employed forms in appendix.

9. Line 375. I interpret h_c7 as SE4. Please specify to facilitate understanding.

10. Line 383, figure 7 caption misspelled.

English language has minor flaws of syntax or spelling.

Round 2

Reviewer 1 Report

The manuscript can be accepted for publication

Author Response

We deeply appreciate your work and suggestions, which are valuable in improving the quality of our manuscript. Should you have any questions about our work, please contact us without hesitate.

Reviewer 2 Report

The manuscript is revised based on the comments and is acceptable in the current form.

Author Response

(The authors gave the same response as above.)

Reviewer 4 Report

Authors have made some modifications to the paper based on the comments from all reviewers. The quality of the presentation has slightly improved. In my opinion, however, the substance of the article has not changed, the implications of the study are quite preliminary and the overall merit is low. Nonetheless, I will credit the recommendation of the other reviewers.

Remaining on the form of the presentation, I see you moved a paragraph of sec.4 into Appendix as I suggested. I do not get why you moved only that parargraph, since that does not really relieve the burden of the section. You should move to appendix all mathematical parts that report a method or a procedure that can be found in the literature, and leave in sec. 4 only those passages, if any, that you originally developed or that report the final equations you used for each method. Otherwise the use of appendix becomes pointless and you can just leave everything in the body of text.

Author Response

Dear reviewer:

    In the last modified version, we moved a state transformation part into the appendix because that this part is a general way to build a standard state equation form which the sliding mode observer is based on.  however, besides that, section 4.1 is our original method to treat health parameters or sensor faults as artificial inputs, and introduce a transformation to create a fictitious output that dimensionally outnumbers the health parameter (or sensor fault) vector, which makes room for the robust design. Section 4.2 to 4.4 is the improved super-twisting sliding mode observer design with coefficient to be adaptive to decrease the chattering. and among which a lot of proof and deduction have already been cited. If all mathematical parts moves to appendix, I'm afraid it will be hardly to understand our method. 

    We deeply appreciate your suggestions, which are valuable in improving the quality of our manuscript. Should you have any questions about our work, please contact us without hesitate.